# Gradient flow dynamics of shallow ReLU networks for square loss and orthogonal inputs

**Etienne Boursier**
TML, EPFL, Switzerland
etienne.boursier@epfl.ch

**Loucas Pillaud-Vivien**
TML, EPFL, Switzerland
loucas.pillaud-vivien@epfl.ch

**Nicolas Flammarion**
TML, EPFL, Switzerland
nicolas.flammarion@epfl.ch

## Abstract

The training of neural networks by gradient descent methods is a cornerstone of the deep learning revolution. Yet, despite some recent progress, a complete theory explaining its success is still missing. This article presents, for orthogonal input vectors, a precise description of the gradient flow dynamics of training one-hidden layer ReLU neural networks for the mean squared error at small initialisation. In this setting, despite non-convexity, we show that the gradient flow converges to zero loss and characterise its implicit bias towards minimum variation norm. Furthermore, some interesting phenomena are highlighted: a quantitative description of the initial alignment phenomenon and a proof that the process follows a specific saddle to saddle dynamics.

## 1 Introduction

Artificial neural networks are nowadays trained successfully to solve a large variety of learning tasks. However, a large number of fundamental questions surround their impressive success. Among them, the convergence to global minima of their non-convex training dynamics and their ability to generalise well despite fitting perfectly the dataset have challenged traditional machine learning belief. While a complete theory is still lacking, the machine learning community has recently come up with key steps that allow to tame the complexity of the problem: proving the convergence of gradient flow to zero loss [Mei et al., 2018, Chizat and Bach, 2018, Sirignano and Spiliopoulos, 2020, Rotskoff and Vanden-Eijnden, 2022], investigating the algorithmic selection of a specific global minimum, often referred as the *implicit bias* of an algorithm [Neyshabur et al., 2014, Zhang et al., 2021]; while paying attention to the importance of the initialisation [Woodworth et al., 2020, Chizat et al., 2019]. The aim of this article is to analyse precisely these three points for regression problems. This is done in a specific setting: for orthogonal inputs, we provide a complete characterisation of the gradient flows dynamics of training one-hidden layer ReLU neural networks with the square loss at small initialisation. We show that this non-convex optimisation dynamics captures most of the complexity mentioned above and thus could be a first step towards analysing more general setups.

**Global convergence of training loss for neural networks.** Showing convergence of the gradient flow to a global minimum is an open and important question. Beyond the lazy regime (see next paragraph), only a few results were proven in the regression setting. The most promising route might be the link with Wassertein gradient flows for infinite neural networks. In that case, global convergence happens under mild conditions [Chizat and Bach, 2018, Wojtowytsch, 2020]. Other works focus on local convergence [Zhou et al., 2021, Safran et al., 2021], or general criteria that eventually fail to encompass practical setups [Chatterjee, 2022, Chen et al., 2022]. These latter works

rest on Polyak-Łojasiewicz inequalities that in fact cannot be satisfied through the whole process if the dynamics travels near saddle points [Liu et al., 2022], as empirically observed [Dauphin et al., 2014]. On the contrary, the present paper proves global convergence without resorting to large overparameterisation, dealing carefully with saddles.

**Feature learning and small initialisation.** The scale of initialisation plays an essential role in the behavior of the training dynamics. Indeed, an important example is that, at large initialisation, known as the *lazy regime* [Chizat et al., 2019], the neurons move relatively slightly implying that the dynamics is nearly convex and described by an effective kernel method with respect to the *Neural Tangent Kernel* [Jacot et al., 2018, Allen-Zhu et al., 2019, Arora et al., 2019]. Instead, we are interested in another regime where the initialisation scale is small. This regime is known to be richer as it performs *feature learning* [Yang and Hu, 2021] but is also more challenging to analyse as it follows a truly non-convex dynamics (see details in Section 4).

**Implicit bias of gradient methods training.** There are many global minima to the mean squared error, i.e. ReLU neural networks that perfectly interpolate the dataset. An important question is to understand which one is selected by the gradient flow for a given initialisation [Neyshabur et al., 2014]. For linear neural networks, this question has been answered thoroughly [Arora et al., 2019, Yun et al., 2021, Min et al., 2021] with a discussion on the role of initialisation [Woodworth et al., 2020] and noise [Pesme et al., 2021]. For non-linear activations such as ReLU, no clear implicit bias criteria have been ever exhibited for the square loss besides a conjecture of a quantisation effect [Maennel et al., 2018]. Finally, note that in the classification setting, the favorable behavior of iterates going to infinity simplifies the analysis to prove implicit biases such as: max-margin for the $\ell_2$ norm in case of linear models [Soudry et al., 2018, Ji and Telgarsky, 2019b], alignment of inner layers for linear neural networks [Ji and Telgarsky, 2019a] and max-margin for the variation norm induced by neural networks [Kurková and Sanguineti, 2001] in the case of one-hidden layer neural networks [Lyu and Li, 2019, Chizat and Bach, 2020].
Beyond the convergence results, the implicit bias characterisation anticipates the generalisation properties of the returned estimate as discussed in Section 3.2.

**Dynamics of training for neural network.** In the regression case, the starting point governs where the flow converges. This observation suggests that a complete analysis of the trajectory may be required when one wants to understand the implicit bias in this case. Such descriptions have been undertaken by Maennel et al. [2018], who describe the initial alignment phase at small initialisation, and Li et al. [2020], Jacot et al. [2021] who conjecture that the dynamics travels from saddle to saddle. These papers provide intuitive content that we prove rigorously in the orthogonal setup.
Finally, closest to our work are the following results on the classification of orthogonally separable data [Phuong and Lampert, 2020, Wang and Pilanci, 2021] and linearly separable, symmetric data [Lyu et al., 2021]. The classification setup provides easier tools to analyse the problem: indeed, after the initial alignment phase, the network has already perfectly classified the data points in these settings. From there, it is known that the training loss converges to zero and that the parameters direction is biased towards KKT points of the max-margin problem [Lyu and Li, 2019, Ji and Telgarsky, 2020]. Such tools cannot be applied after the alignment phase for regression, and we resort to a refined analysis of the trajectory to show both global convergence and implicit bias. On the other hand, Lyu et al. [2021] require a precise description of the dynamics to ensure convergence towards specific KKT points of the max-margin problem. Yet, the analysis of the dynamics is simplified by their symmetry assumption: the trajectory does not go through intermediate saddles and all the labels are simultaneously fitted. On the contrary, the dynamics we describe travels near an intermediate saddle point which separates two distinct fitting phases. This behaviour largely complicates the analysis, besides being more representative of the saddle to saddle dynamics observed in general settings.

## 1.1 Main contributions

We make the following contributions.

- We prove the convergence of the gradient flow towards a global minimum of the non-convex training loss for small enough initialisation and finite width.
- We characterise the global optimum retrieved for infinitesimal initialisation as a minimum $\ell_2$ norm interpolator, which implies a minimum variation norm in terms of prediction function.
- As important as the convergence result, the dynamics is portrayed in Section 4: we quantitatively detail its different phases (alignment and fitting) and show it follows a saddle to saddle dynamics.

## 1.2 Notations

We denote by $\mathbb{1}_A$ the function equal to 1 if $A$ is true and 0 otherwise. $\mathcal{U}(S)$ is the uniform distribution over the set $S$ and $\mathcal{N}(\mu, \Sigma)$ is a Gaussian of mean $\mu$ and covariance $\Sigma$. We denote $\nabla_\theta h_\theta(x)$ the gradient of $\theta \mapsto h_\theta(x)$ at fixed $x$. For any $n \in \mathbb{N}^*$, $[\![n]\!]$ denotes the tuple of integers between 1 and $n$. The scalar product between $x, y \in \mathbb{R}^d$ is denoted by $\langle x, y \rangle$ and the Euclidean norm is denoted by $\|\cdot\|$ and called $\ell_2$. $\mathcal{S}_{d-1}$ denotes the sphere of $\mathbb{R}^d$ for the Euclidean norm. $B(\theta, r)$ is the Euclidean ball of center $\theta$ and radius $r$. All the detailed proofs of the claimed results are deferred to the Appendix.

# 2 Setup and preliminaries

## 2.1 One-hidden layer neural network and training loss

**Model.** Let us fix an integer $n \in \mathbb{N}^*$ as well as input data $(x_1, \ldots, x_n) \in (\mathbb{R}^d)^n$ and outputs $(y_1, \ldots, y_n) \in \mathbb{R}^n$. We are interested in the minimisation of the mean squared error:

$$L(\theta) := \frac{1}{2n} \sum_{k=1}^n (h_\theta(x_k) - y_k)^2, \qquad \text{where } h_\theta(x) := \sum_{j=1}^m a_j \sigma(\langle w_j, x \rangle) \tag{1}$$

is a one-hidden layer neural network of width $m$ defined with parameters $\theta = (a, W) \in \mathbb{R}^m \times \mathbb{R}^{m \times d}$. The vector $a \in \mathbb{R}^m$ stands for the weights of the last layer and $W^\top = [w_1 \cdots w_m] \in \mathbb{R}^{d \times m}$, where each $w_j \in \mathbb{R}^d$ represents a hidden neuron. To encompass the effect of the bias, an additional component can be added to the inputs $x^\top \leftarrow [x^\top, 1]$ without changing our results. Finally, the activation function $\sigma$ is the ReLU: $\sigma(x) := \max\{0, x\}$.

We introduce here the main assumptions on the data inputs.

**Assumption 1.** *The input points form an orthonormal family, i.e. $\forall k, k' \in [\![n]\!]$, $\langle x_k, x_{k'} \rangle = \mathbb{1}_{k=k'}$.*

The data are assumed to be normalized only for convenience—the real limitation being that they are pairwise orthogonal. This assumption is exhaustively discussed in Section 3.2.

**Assumption 2.** *For all $k \in [\![n]\!]$, $y_k \neq 0$ and $\sum_{k|y_k>0} y_k^2 \neq \sum_{k|y_k<0} y_k^2$.*

This assumption on the data output is mild, e.g. has zero Lebesgue measure, and only permits to exclude degenerate situations.

**Gradient flow.** As the limiting dynamics of the (stochastic) gradient descent with infinitesimal step-sizes [Li et al., 2019], we study the following gradient flow

$$\frac{\mathrm{d}\theta^t}{\mathrm{d}t} = -\nabla L(\theta^t) = -\frac{1}{n} \sum_{k=1}^n (h_{\theta^t}(x_k) - y_k) \nabla_\theta h_{\theta^t}(x_k), \tag{2}$$

initialised at $\theta^0 := (a^0, W^0)$. Since the ReLU is not differentiable at 0, the dynamics should be defined as a subgradient inclusion flow [Bolte et al., 2010]. However, we show in Appendix D that the *only* ReLU subgradient that guarantees the existence of a global solution is $\sigma'(x) = \mathbb{1}_{x>0}$. Hence, we stick with this choice throughout the paper. Another important difficulty of this non-differentiability is that Cauchy-Lipschitz theorem does not apply and uniqueness is not ensured. There have been attempts to define the solution of this Ordinary Diffential Equation (ODE) unequivocally [Eberle et al., 2021] as well as ways to circumvent this difficulty by resorting to smooth activations or additional data assumptions [Wojtowytsch, 2020, Chizat and Bach, 2020]. Yet, we do not follow this line and demonstrate our results *for all the gradient flows* satisfying Equation (2).

## 2.2 Preliminary properties and initialisation

Let us derive here some preliminary properties of the gradient flows. If we rewrite explicitly the dynamics of Equation (2) on each layer separately, we have straightforwardly that for all $j \in [\![m]\!]$,

$$\frac{\mathrm{d}a_j^t}{\mathrm{d}t} = \langle D_j^{\theta^t}, w_j^t \rangle \qquad \text{and} \qquad \frac{\mathrm{d}w_j^t}{\mathrm{d}t} = D_j^{\theta^t} a_j^t, \tag{3}$$

where $D_j^{\theta^t} := -\frac{1}{n}\sum_{k=1}^n \mathbb{1}_{\langle w_j^t, x_k\rangle > 0}\left(h_{\theta^t}(x_k) - y_k\right)x_k$ is a vector of $\mathbb{R}^d$. This vector solely depends on $\theta^t$ through the prediction function $h_{\theta^t}$ and on the neuron $j$ through its activation vector $\mathsf{A}(w_j^t)$; where, for a vector $w \in \mathbb{R}^d$, $\mathsf{A}(w) := (\mathbb{1}_{\langle w, x_1\rangle > 0}, \dots, \mathbb{1}_{\langle w, x_n\rangle > 0}) \in \{0, 1\}^n$. From Equation (3), we deduce the following balancedness property [Arora et al., 2019].

**Lemma 1.** *For all $t \geq 0$ and all $j \in [\![m]\!]$, $(a_j^t)^2 - \|w_j^t\|^2 = (a_j^0)^2 - \|w_j^0\|^2$. Assume furthermore that for all $j \in [\![m]\!]$, the initialisation is balanced and non-zero: $|a_j^0| = \|w_j^0\| > 0$. Then $|a_j^t| = \|w_j^t\| > 0$ and letting $\mathsf{s} = \mathrm{sign}(a^0) \in \{1, -1\}^m$, for all $t \geq 0$, we have that $a_j^t = \mathsf{s}_j\|w_j^t\|$.*

Importantly, by Lemma 1, the study of Equation (3) reduces to the hidden layer $W$ solely. We consider the following balanced initialisation:

$$\theta^0 = (a^0, W^0) \quad \text{with} \quad \begin{cases} w_j^0 = \lambda\, g_j \text{ where } g_j \overset{\text{i.i.d.}}{\sim} \mathcal{N}(0, I_d), \\ a_j^0 = \mathsf{s}_j\|w_j^0\| \text{ where } \mathsf{s}_j \overset{\text{i.i.d.}}{\sim} \mathcal{U}(\{-1, 1\}). \end{cases} \tag{4}$$

As already stated, we are interested in the regime where the initialisation scale $\lambda > 0$ is small. We also introduce the following sets of neurons that are crucial in the fitting process

$$S_{+,1} := \left\{j \in [\![m]\!] \mid \mathsf{s}_j = +1 \quad \text{and} \quad \text{for all } k \text{ such that } y_k > 0,\ \langle w_j^0, x_k\rangle \geq 0\right\}, \tag{5}$$

$$S_{-,1} := \left\{j \in [\![m]\!] \mid \mathsf{s}_j = -1 \quad \text{and} \quad \text{for all } k \text{ such that } y_k < 0,\ \langle w_j^0, x_k\rangle \geq 0\right\}. \tag{6}$$

**Assumption 3.** *The sets $S_{+,1}$ and $S_{-,1}$ are both non-empty.*

Assumption 3 states that there are some neurons in two given cones at initialisation. It holds with probability 1 when the support of initialisation covers all directions and the width $m$ of the network goes to infinity. This is thus a weaker condition than the *omni-directionality* of neurons at initialisation [Wojtowytsch, 2020], which is instrumental to show convergence in the mean field regime [Chizat and Bach, 2018]. On the other hand, it is stronger than the alignment condition of [Abbe et al., 2022], which is known to be necessary for weak learning but might not lead to the implicit bias described in the next section.

## 3 Convergence and implicit bias characterisation

### 3.1 Main result

Theorem 1 below states our main result on the convergence and implicit bias of one-hidden layer ReLU networks for regression tasks with orthogonal data.

**Theorem 1.** *Under Assumptions 1 to 3, there exists $\lambda^* > 0$ depending only on the data and the width such that, if $\lambda \leq \lambda^*$, the gradient flow initialised according to Equation (4) converges almost surely to some $\theta_\lambda^\infty$ of zero training loss, i.e. $L(\theta_\lambda^\infty) = 0$. Furthermore, there exists $\theta^*$ such that*

$$\lim_{\lambda \to 0}\lim_{t \to \infty}\theta^t = \theta^* \in \underset{L(\theta)=0}{\mathrm{argmin}}\ \|\theta\|^2. \tag{7}$$

The significance of this result is thoroughly discussed in Section 3.2. Note that a quantitative and non-asymptotic version of Theorem 1, both in time and $\lambda$, is stated in Lemma 12 (Appendix B). Roughly, it states that the dynamics has already nearly converged after a time of order $-\ln(\lambda)$ and then that the convergence happens at exponential speed. Note also that the neural network need not be overparametrised for the result to hold: the only sufficient and necessary requirement on the width $m$ stems from Assumption 3.

**Sketch of proof.** The proof of Theorem 1 rests on a precise description of the training dynamics, which is divided into four different phases. We here only sketch it at a very high level and a more thorough description, with quantitative intermediate lemmas, is given in Section 4.

During the first phase, hidden neurons align to a few representative directions, while remaining close to 0 in norm. In particular, all hidden neurons in $S_{+,1}$ (resp. $S_{-,1}$) align with some key vector $D_+$ (resp. $-D_-$) defined in Section 4.1. During the second phase, the neurons aligned with $D_+$ grow in norm, while staying aligned with $D_+$, until fitting all the positive labels of the dataset (up to some error scaling with $\lambda$). Meanwhile, all the other neurons stay idle. Then similarly, the neurons aligned

with $-D_-$ grow in norm during the third phase, until nearly fitting all the negative labels. Meanwhile, these neurons remain aligned with $-D_-$ and all other neurons remain idle. The precise description of these three phases is obtained by analysing the solutions of the limit ODEs when $\lambda = 0$. The approximation errors that occur from dealing with non-zero $\lambda$ are then carefully handled via Grönwall comparison arguments. Due to the large time scales (of order $-\ln(\lambda)$), the error can propagate on such large time spans. Handling these error terms is the main challenge of our proof and remains intricate despite the orthogonality assumption.

After these three phases (which last a time $-\ln(\lambda)/\|D_-\|$), the parameters vector is close to some minimal $\ell_2$-norm interpolator. From there we show, exploiting a local Polyak-Łojasiewicz condition, that the dynamics converges at exponential speed to a global minimum close to this interpolator.

### 3.2 Discussion

Even if the orthogonal setting we consider is quite restrictive, it carries several characteristics that may be generic, either because they have been observed empirically, shown in related contexts or simply conjectured. We discuss these important points below.

**Convergence to zero loss.**    Theorem 1 states that the gradient flow converges to zero loss. Such a result is simple to show when the loss satisfies a Polyak-Łojasiewicz (PL) inequality [Bolte et al., 2007]: $\|\nabla L\|^2 \geq cL$ for $c > 0$. However, here, as the dynamics travels near saddles, this inequality is not verified through all the process. Circumventing this global argument, it is yet possible to formulate a refined analysis and show convergence if the dynamics arrives in a region where a local PL stands with a large enough constant. This refined analysis, inspired by the recent work of Chatterjee [2022][1], allows to characterise properly the last phase of the dynamics. We believe that this approach may help in showing convergence in other non-convex gradient flow/descent.

**On the implicit bias.**    Additionally, Theorem 1 states that the gradient flow at infinitesimally small initialisation *selects* global minimisers with the smallest $\ell_2$ parameter norm. To our knowledge, this is the first characterisation of the implicit bias for regression with non-linear neural networks. Although it might not hold for some degenerate situations [Vardi and Shamir, 2021], we believe it to be true beyond the orthogonal case.

This regularisation is *implicit*, meaning that this effect does not result from any explicit regularisation (e.g. weight decay) performed during training [Shevchenko et al., 2021, Parhi and Nowak, 2022]. This is only a consequence of the inner structure of the gradient flow and the scale of initialisation.

Furthermore, the implicit bias in parameter space can be translated in function space. Indeed, if we introduce formally the space of (infinite) neural networks, i.e. functions written as $f(x) := \int \sigma(\langle\theta, x\rangle)\mathrm{d}\mu(\theta)$, where $\mu$ is a signed finite measure on $\mathcal{S}_{d-1}$. Then, we can define the *variation norm*, $\|f\|_{\mathcal{F}_1}$, as the infimum of $|\mu|(\mathcal{S}_{d-1})$ over such representations [Kurková and Sanguineti, 2001, Bach, 2017]. We have the following link between the two formulations

$$\min_{L(\theta)=0} \frac{1}{2}\|\theta\|_2^2 = \min_{L(f)=0} \|f\|_{\mathcal{F}_1}, \tag{8}$$

with a slight abuse of notation when defining $L(f)$. Note that the result in terms of the $\ell_2$-norm of the parameters is strictly stronger than that of the $\mathcal{F}_1$-norm of the function [Neyshabur et al., 2014].

Following Equation (8), note the striking parallel between the inductive bias of infinitesimally small initialisation for regression and that of the classification problem with the logistic loss as a max-margin problem with respect to the $\mathcal{F}_1$-norm [Chizat and Bach, 2020]. As already observed in the linear case [Woodworth et al., 2020], in contrast with classification, infinitesimally small initialisation is instrumental in regression to be biased towards small $\mathcal{F}_1$-norm functions. The role of initialisation is illustrated empirically in Appendix A.

Finally, let us stress that we did not address the question of what functions solve Equation (8), nor the question of the generalisation implied by such a bias. Related works on the first point come from a functional description of norms related to $\mathcal{F}_1$ [Savarese et al., 2019, Ongie et al., 2019, Debarre et al., 2022]. For the generalisation properties of small $\mathcal{F}_1$ norm functions, we refer to Kurková and Sanguineti [2001], Bach [2017]. Importantly, we recall that the question of how well low $\mathcal{F}_1$-norm functions generalise depends heavily on the *a priori* we have on the ground-truth [Petrini et al., 2022].

---

[1]Note that the argument is certainly not new, but the cited article has the benefit of clearly presenting it.

**The initial alignment phenomenon.** An important characteristics of the loss landscape is that the origin is a saddle point. Hence, as the dynamics is initialised at small scale $\lambda$, the radial movement is slow and neurons move out of the saddle after time scale $-\ln \lambda$. Meanwhile, the tangential movement of the neurons rules the dynamics and aligns their directions towards specific vectors. This has been first explained by Maennel et al. [2018] and referred as the *quantisation* phenomenon, because neural networks weights collapse to a small finite number of directions. We emphasise that this phase happens generically when initialisation is near the origin and that this part of our analysis can be directly extended to the general (i.e. non-orthognal) case. Phuong and Lampert [2020], Lyu et al. [2021] analysed a similar early alignment for classification with specific data structures.

**The saddle to saddle dynamics.** When initialising the dynamics of a gradient flow near a saddle point of the loss, it is expected (but hard to prove generically) that the dynamics will alternate slow movements near saddles and rapid junctions between them. Such a behavior has been conjectured for linear neural networks [Li et al., 2020, Jacot et al., 2021] initialised near the origin. We precisely prove that such a phenomenon occurs: after initialisation, the dynamics visits one strict saddle. See Section 4, Fact 1 for more details.

**Limitations and possible relaxations.** As its main limitation, Theorem 1 assumes orthogonal data points $x_k$. The orthogonality assumption disentangles the analysis as the different phases, where either the neurons align towards some direction or grow in norm, are well separated in that case. More precisely, the neurons do not change in direction once they have a non-zero norm in the case of orthogonal data. This separation between alignment and norm growth does not hold in the general case, as observed empirically in Appendix A. Extending our result to more general data thus remains a major challenge and requires additional theoretical tools. Nonetheless, as it can be the case in high dimension, our analysis can easily be extended to nearly orthogonal data where $|\langle x_k, x_{k'} \rangle| \leq \delta$, with $\delta$ of order $\lambda$. If however $\delta$ is much larger than the initialisation scale, the dynamics is drastically different and becomes as hard as the general case to analyse. In Appendix A, we observe similar dynamics for high dimensional data, where the loss converges towards 0, goes through an intermediate saddle point and the final solution is close to a 2 neurons network.

A minor assumption is the balanced initialisation, i.e. $\|w_j^0\| = |a_j^0|$. If instead we initialise $a^0$ as a Gaussian scaling with $\lambda$, the initialisation would be nearly balanced for small $\lambda$. This assumption is thus mostly used for simplicity and our analysis can be extended to unbalanced initialisations.

It is unclear whether our analysis can be extended to any homogeneous activation function. The training trajectory might indeed not be biased towards minimal $\ell_2$-norm for *leaky ReLU* activations [Lyu et al., 2021, Theorem 6.2]. Contrary to some beliefs, it suggests that the $\ell_2$ implicit bias phenomenon does not occur for any homogeneous activation function, but might instead be specific to the ReLU.

**The overparameterisation regime.** Assumption 3 states a deterministic condition to guarantee convergence towards a minimal norm interpolator. This condition is not only sufficient, but also *necessary* for implicit bias towards minimal $\ell_2$ norm. For isotropic initialisations, the width $m$ needs to be exponential in the number of data points $n$ for Assumption 3 to hold with high probability.

With a smaller (e.g. polynomial in $n$) number of neurons, Assumption 3 does not hold anymore. In that case, the training loss should still converge to 0, but the $\ell_2$-norm of the parameters will not be minimal. More precisely, the estimated function will have more than two kinks. However, an adapted analysis might still show some sparsity in the number of kinks (and thus a weak bias) of the final solution. The training trajectory would then go through multiple saddle points (one saddle per kink).

**Scale of initialisation.** The exact value of $\lambda^*$ is omitted for exposition's clarity. Roughly, it can be inferred from the analysis that $\lambda^*$ scales as $\frac{\Theta(1)}{\sqrt{m}} e^{-\Theta(n)}$. Interestingly, the $\frac{1}{\sqrt{m}}$ term is reminiscent of the mean field regime, which is known to induce implicit bias [Chizat and Bach, 2020, Lyu et al., 2021]. On the other hand, the exponential dependency in $n$ is common in the implicit bias literature [Woodworth et al., 2020]. For larger values of $\lambda$ (but still in the mean field regime), the parameters empirically seem to also converge towards a minimal norm interpolator. The analysis yet becomes more intricate and we do not observe any separation between Phase 2 and Phase 3, i.e. there is no intermediate saddle in the trajectory.

# 4 Fine dynamics description: alignments and saddles

This section describes thoroughly the training dynamics of the gradient flow. It presents and discusses quantitative lemmas on the state of the neural network at the end of each different phase. In particular, mathematical formulations of the early alignment and saddle to saddle phenomena are provided.

## 4.1 Additional notations

First, we need to introduce additional notations for this section. We define vectors $D_+$ and $D_-$ that are the two directions towards which the neurons align

$$D_+ := \frac{1}{n} \sum_{k|y_k>0} y_k x_k \qquad \text{and} \qquad D_- := \frac{1}{n} \sum_{k|y_k<0} y_k x_k.$$

We also need to define $c := \max_{j \in [\![m]\!]} \|w_j^0\|/\lambda$ and $r := \|D_+\|/\|D_-\|$. Assumption 2 implies that $r \neq 1$, and by symmetry we can assume $r > 1$ without any loss of generality. We additionally fix constants $\lambda_*, \varepsilon > 0$, small enough and depending only on the dataset and the width $m$.

**Spherical coordinates.** As radial and tangential movements are almost decoupled during the dynamics, it is natural to introduce the spherical coordinates of the neurons: for all $j \in [\![m]\!]$, denote $w_j = e^{\rho_j} \cdot \mathsf{w}_j$, where $\rho_j = \ln \|w_j\| \in \mathbb{R}$ and $\mathsf{w}_j = w_j/\|w_j\| \in \mathcal{S}_{d-1}$. In these adapted coordinates, the system of ODEs (3) reduces to:

$$\frac{d\rho_j^t}{dt} = \mathsf{s}_j \langle D_j^{\theta^t}, \mathsf{w}_j^t \rangle \qquad \text{and} \qquad \frac{d\mathsf{w}_j^t}{dt} = \mathsf{s}_j \left( D_j^{\theta^t} - \langle D_j^{\theta^t}, \mathsf{w}_j^t \rangle \mathsf{w}_j^t \right). \tag{9}$$

## 4.2 Training dynamics

This section precisely describes the phases of the dynamics, summarised in Figure 1:

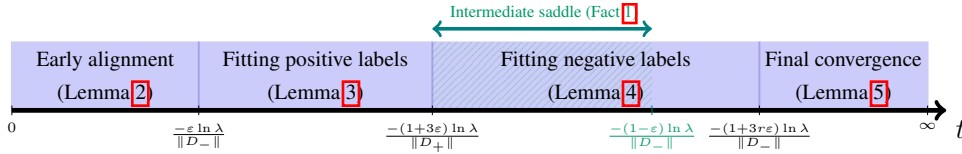

**Figure 1:** Timeline of the training dynamics.

**Neuron alignment phase.** During the first phase, all the neurons remain small in norm, while moving tangentially (i.e. in directions). The neurons align according to several key directions: an initial clustering of neurons' directions happens in this early phase, as observed by Maennel et al. [2018]. As the neurons have small norm, $h_{\theta^t} \approx 0$ for this phase and Equation (9) approximates

$$\frac{d\mathsf{w}_j^t}{dt} \approx \mathsf{s}_j \left( D_j^0 - \langle D_j^0, \mathsf{w}_j^t \rangle \mathsf{w}_j^t \right). \tag{10}$$

This ODE corresponds to the descent/ascent gradient flow (depending on the sign of $\mathsf{s}_j$) on the sphere with objective $\langle D_j^0, \mathsf{w}_j \rangle$. All neurons end up minimizing or maximizing their scalar product with $D_j^0$, which only depends on the activation of $\mathsf{w}_j$. As a consequence, neurons with similar activations align towards the same vector, leading to some quantisation of the neurons' directions. This alignment happens in a relatively short time, so that the neurons cannot largely grow in norm. Lemma 2 below quantifies this effect for neurons in $S_{+,1}$ and $S_{-,1}$, which are crucial to the training dynamics. Since all other neurons remain small in norm during the whole process, we do not focus on their direction.

**Lemma 2** (First phase). *For $\lambda \leq \lambda^*$, we have the following inequalities for $t_1 = \frac{-\varepsilon \ln(\lambda)}{\|D_-\|}$:*

(i) *neurons in $S_{+,1}$ are aligned with $D_+$:*   $\forall j \in S_{+,1}, \langle \mathsf{w}_j^{t_1}, D_+ \rangle \geq (1 - 2\lambda^\varepsilon)\|D_+\|$,

(ii) *neurons in $S_{-,1}$ are aligned with $-D_-$:*   $\forall j \in S_{-,1}, \langle \mathsf{w}_j^{t_1}, -D_- \rangle \geq (1 - 2\lambda^\varepsilon)\|D_-\|$,

(iii) *all neurons have small norm:*   $\forall j \in [\![m]\!], \|w_j^{t_1}\| \leq 2c\lambda^{1-r\varepsilon}$.

**Fitting positive labels.** During the second phase, the norm of the neurons in $S_{+,1}$ (which are aligned with $D_+$) grows until fitting all positive labels. Meanwhile, all the other neurons do not move significantly. The key approximate ODE of this phase is given for $u_+(t) := \sum_{j \in S_{+,1}} \|w_j^t\|^2$ by

$$\frac{\mathrm{d}u_+(t)}{\mathrm{d}t} \approx 2\|D_+\| \left(1 - \frac{u_+(t)}{n\|D_+\|}\right) u_+(t).$$

This equation implies that $u_+(t)$, the sum of the squared norms of neurons in $S_{+,1}$, eventually converges to $n\|D_+\|$ within a time $-\ln(\lambda)/\|D_+\|$ . Meanwhile, it needs to be shown that these neurons remain aligned with $D_+$ and that the other neurons remain small in norm. This fine control is technical and relies on the orthogonality assumption. If data were not orthogonal, neurons could indeed realign while growing in norm as illustrated by Figure 4d in Appendix A. Lemma 3 below describes the state of the network at the end of the second phase.

**Lemma 3** (Second phase)**.** *If $\lambda \leq \lambda^*$, then for some time $t_2 \leq -\frac{1+3\varepsilon}{\|D_+\|} \ln(\lambda)$:*

   *(i) neurons in $S_{+,1}$ are aligned with $D_+$:*   $\forall j \in S_{+,1}, \langle \mathsf{w}_j^{t_2}, D_+ \rangle \geq \|D_+\| - \lambda^{\frac{\varepsilon}{2}}$,

   *(ii) neurons in $S_{+,1}$ have a large norm:*   $\sum_{j \in S_{+,1}} \|w_j^{t_2}\|^2 = n\|D_+\| - \lambda^{\frac{\varepsilon}{5}}$,

   *(iii) other neurons have small norm:*   $\forall j \in [\![m]\!] \setminus S_{+,1}, \|w_j^{t_2}\| \leq 2c\lambda^\varepsilon$.

These three points directly imply that the loss is of order $\lambda^{\frac{\varepsilon}{5}}$ on the positive labels at time $t_2$.

**Saddle to saddle dynamics.** As explained above, the positive labels are almost fitted by the action of the neurons belonging to $S_{+,1}$ at the end of the second phase, whereas the other neurons still have infinitesimally small norm. At this point, the dynamics has reached the vicinity of a strict saddle point and requires a long time to escape it. The analysis actually leads to the following fact:

**Fact 1.** *There exists a (strict) saddle point $\theta_S \neq 0$ of $L$ such that if $\lambda \leq \lambda^*$:*

$$\forall t \in \left[-\frac{1+3\varepsilon}{\|D_+\|} \ln(\lambda), -\frac{1-\varepsilon}{\|D_-\|} \ln(\lambda)\right], \quad \text{we have } \|\theta^t - \theta_S\| \leq \lambda^{\frac{\varepsilon}{5}}.$$

The training trajectory thus starts at the saddle point $0$ and passes through a second non-trivial saddle point at the end of the second phase. This lemma illustrates the phenomenon of *saddle to saddle dynamics* discussed in Section 3.2 and conjectured for linear models by Li et al. [2020], Jacot et al. [2021]. This intermediate saddle point is escaped when the norms of the neurons in $S_{-,1}$ have significantly grown (i.e. become non-zero), which happens during a third phase described below.

**Fitting negative labels.** The norm of the neurons in $S_{-,1}$ (which are aligned with $-D_-$) grows until fitting all negative labels during the third phase. Meanwhile, all other neurons do not move significantly. The additional difficulty in the analysis of this phase compared to the second one is that of controlling the possible movements of neurons in $S_{+,1}$. Their norm is indeed large during the whole phase, but they do not change consequently, because the positive labels are nearly perfectly fitted.

**Lemma 4** (Third phase)**.** *If $\lambda \leq \lambda^*$, then for some time $t_3 \leq -\frac{1+3r\varepsilon}{\|D_-\|} \ln(\lambda)$:*

   *(i) neurons in $S_{-,1}$ are aligned with $-D_-$:*   $\forall j \in S_{-,1}, \langle \mathsf{w}_j^{t_3}, -D_- \rangle \geq \|D_-\| - \lambda^{\frac{\varepsilon}{14}}$,

   *(ii) neurons in $S_{-,1}$ have a large norm:*   $\sum_{j \in S_{-,1}} \|w_j^{t_3}\|^2 = n\|D_-\| - \lambda^{\frac{\varepsilon}{29}}$,

   *(iii) neurons in $S_{+,1}$ did not move since phase 2:*   $\forall j \in S_{+,1}, \|w_j^{t_2} - w_j^{t_3}\| \leq \lambda^{\frac{\varepsilon}{15}}$,

   *(iv) other neurons have small norm:*   $\forall j \in [\![m]\!] \setminus (S_{+,1} \cup S_{-,1}), \|w_j^{t_3}\| \leq 3c\lambda^\varepsilon$.

Thanks to the orthogonality assumption, the set of minimal $\ell_2$-norm interpolators can be exactly described by Proposition 1 in Appendix C. The minimal interpolators are actually *equivalent* to a neural network of width 2: the first hidden neuron is collinear with $D_+$ and the second one is collinear with $-D_-$. Lemma 4 then ensures at the end of the third phase that the parameter vector is $\lambda^{\frac{\varepsilon}{29}}$-close to an interpolator $\theta^*$ of minimal $\ell_2$-norm, that does not depend on $\lambda$ (see Lemma 12). It remains to show that the training trajectory converges to a point close to this interpolator at infinity.

**Convergence phase.** To show this final convergence, we use a local PL condition given by Lemma 5.

**Lemma 5** (Local PL condition)**.** *For $\lambda \leq \lambda^*$, we have the following lower bound on the PL constant*

$$\inf_{\theta \in B(\theta^*, \lambda^{\frac{\varepsilon}{240}}) \cap \Theta} \frac{\|\nabla L(\theta)\|^2}{L(\theta)} \geq \|D_-\|,$$

*where $\Theta$ is the set of parameters verifying the balancedness property.*

Adapting arguments from the recent work by Chatterjee [2022], this implies that the training trajectory converges to an interpolator and stays in the aforementioned ball. It thus converges exponentially at a rate $\|D_-\|$ to a point close to a minimal norm interpolator, and the distance to this point goes to $0$ when $\lambda$ goes to $0$, hence implying Theorem 1. This exponential rate is only asymptotical: the dynamics still require a large time $-\ln(\lambda)/\|D_-\|$ to escape the two first saddles.

**Remark 1** (Local PL in the ReLU case). *Note that, strictly speaking, Theorem 1.1 of Chatterjee [2022] cannot be applied directly to our setup because (i) the infimum is taken on the intersection of a ball and the set $\Theta$ of balanced weights, (ii) the ReLU and hence the loss are not $C^2$ as required. We thank Spencer Frei for bringing us this point after the paper acceptance. Yet, the reformulation of Chatterjee [2022] in our case is straightforward and does not provide any more insight. To avoid confusion, we decide to omit the proof here and postpone its writing in an independent note for later. This will also be the occasion to put emphasis on the fact that, in some cases, such a result can be applied in the ReLU case without much difficulty.*

## 5   Experiments

This section confirms empirically the dynamics described in Section 4 on an orthogonal toy example. The code and animated versions of the figures are available in `github.com/eboursier/GFdynamics`. Additional experiments can be found in Appendix A; they illustrate the necessity of small initialisation for implicit bias and present similar experiments on non-orthogonal toy data. For the latter, we observe some similar training phenomena, but major differences appearing in the dynamics highlight the difficulty of dealing with non-orthogonality.

We consider the following two-point dataset: $x_1, y_1 = (-0.5, 1), -1$ and $x_2, y_2 = (2, 1), 1$. It corresponds to unidimensional data with a second $1$ coordinate for the bias term. We choose unidimensional data for a simpler visualisation. However, it restricts the number of observations to $n = 2$ to maintain orthogonality. Also, the inputs' norms are not $1$ here, but we recall that our analysis is not specific to this case. The width of the neural network is $m = 60$. We choose a balanced initialisation at scale $\lambda = 10^{-6}/\sqrt{m}$. We then run gradient descent with a step size $10^{-3}$ to approximate the gradient flow trajectory.

Figure 2 shows the training dynamics on this example. In particular, the state of the network is shown at different steps. In Figure 2a, all the neurons are close to $0$ at initialisation. Figure 2b shows the end of the first phase, where the neurons are aligned towards two key directions. After the second phase, shown in Figure 2c, all the neurons aligned with $D_+$ have grown in norm and the positive label is perfectly fitted. Similarly at the end of third phase in Figure 2d, all neurons aligned with $-D_-$ have grown in norm and the negative label is fitted.

At the end of training, the loss is $0$ and the estimated function is *simple*. In particular, it only has two kinks, which illustrates the sparsity induced by the implicit bias. Also, the final estimated function might be counter-intuitive. Previous works on implicit bias indeed conjectured that the learned estimator is linear if the data can be linearly fitted [Kalimeris et al., 2019, Lyu et al., 2021]. However, the learned function in Figure 2d has a smaller $\mathcal{F}_1$-norm than the linear interpolator.

Figure 3 shows the evolution of the loss during training. The saddle to saddle dynamics is well observed here: the parameters vector starts from the $0$ saddle point at initialisation and needs $5000$ iterations to leave this first saddle. A second saddle is then encountered at the end of the second phase and the trajectory only leaves this saddle around iteration $11000$, once the norm of the neurons in $S_{-,1}$ start being significant during the third phase. All these different experiments confirm Theorem 1 and the precise dynamics described in Section 4. Moreover, such training phenomena are not specific to the orthogonal data case, as observed in Appendix A.

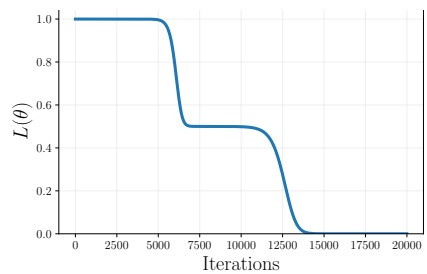

**Figure 3:** Evolution of the training loss.

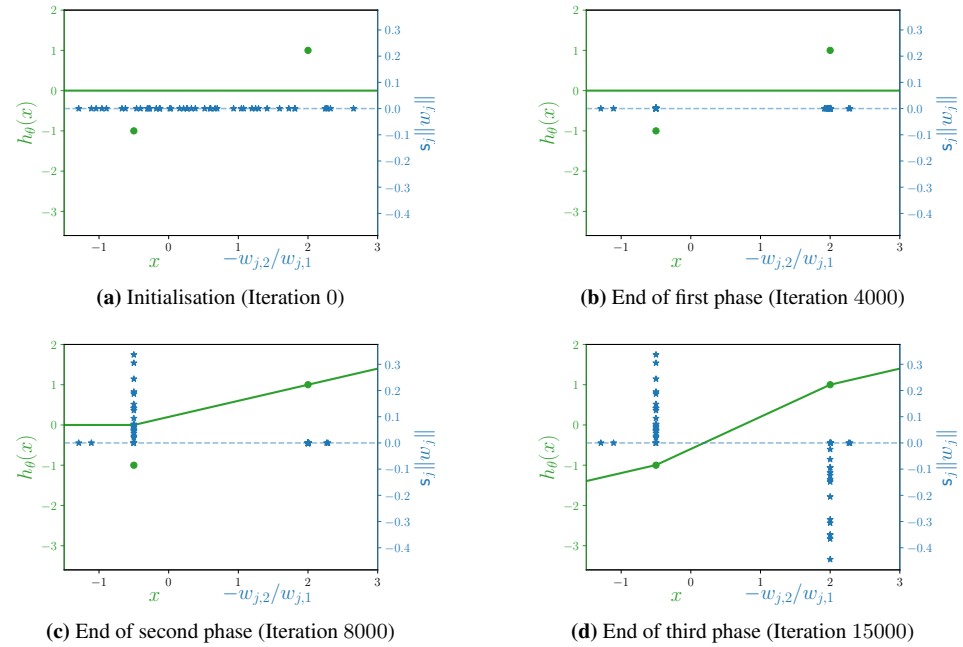

**(a)** Initialisation (Iteration 0)

**(b)** End of first phase (Iteration 4000)

**(c)** End of second phase (Iteration 8000)

**(d)** End of third phase (Iteration 15000)

**Figure 2:** State of training at different stages. The green dots correspond to the data, while the green line is the estimated function $h_\theta$. Each blue star represents a neuron $w_j$: its $x$-axis value is given by $-w_{j,2}/w_{j,1}$, which coincides with the position of the kink of its associated ReLU; its $y$-axis value is given by $\mathsf{s}_j\|w_j\|$, which we recall is the associated value of the output layer.

## 6    Conclusion and perspectives

We have shown that the training of non-linear neural networks on orthogonal data presents a rich dynamics with a small and omnidirectional intialisation. Convergence holds generically despite a truly non-convex landscape and the limit enjoys an implicit bias as a minimum $\ell_2$ parameter norm. Obviously, removing the orthogonal assumption on the inputs, while keeping a fine level of description is a major, but difficult, perspective for future work. Another key point to better understand the good generalisation of neural networks is to analyse the properties of the functions solving the minimum variation norm problem stated in Equation (8).

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
