# OpenReview forum: "Gradient flow dynamics of shallow ReLU networks for square loss and orthogonal inputs"
_NeurIPS.cc/2022/Conference — NeurIPS 2022 Accept_

### Official Review · Reviewer_4xwj · 2022-07-10

**Rating:** 7
**Confidence:** 5
**Soundness:** 3 good
**Presentation:** 3 good
**Contribution:** 3 good

**Summary:**

The paper studies gradient flow dynamics of shallow ReLU networks for regression with square loss. Authors consider a setting of orthogonal inputs in the regime of vanishingly small initialization. In this setting, they provide a characterization of stages of gradient flow on square loss (see Fig. 1), which allows them to draw a few qualitative results about training dynamics: convergence to nearly min-norm interpolator, justification of initial alignment phenomenon and “saddle-to-saddle” behavior of dynamics. Notably, the analysis does not exploit overparameterization and a main technical assumption (A3) (except the data orthogonality) on existence of neurons for which input and last layer weight agree on sign with the label is a mild one (or at least weaker than the one considered previously in related to zero-loss convergence papers).

**Questions:**

I suggest that authors add a few missing relevant references accompanied by discussion:
- “A type of generalization error induced by initialization in deep neural networks” by Yaoyu Zhang et al., 2020 - shows convergence to the closest to init interpolator for lazy training regime (which in some cases might result in selecting the min-norm one)
- There are a few references on generalization capabilities of “min-norm” interpolators (although mostly for linear models), e.g. “Tight bounds for minimum l1-norm interpolation of noisy data” by Guillaume Wang et al., 2022, which would be helpful to add for completeness in corresponding discussion.
- “Gradient dynamics of shallow univariate ReLU networks” by Williams et al., 2019 - shows implicit bias in case of univariate regression depending on the init scale.

Authors should be a bit more explicit about stating that the result holds for a particular sequence (lambda->0) and that at any non-zero lambda dynamics approach but not exactly min-norm, and that the selected min-norm solution might not be unique, and that there is no description (as far as I understood) of a particular min-norm solution selected (e.g. if it is the closest to init?) . Although some of these points can be inferred from statements alone, I would suggest enhancing them for a broader audience.


**Limitations:**

Authors thoroughly discuss the limitations of the proposed approach.

**Strengths And Weaknesses:**

Strengths:
- Precise characterization of training dynamics under “non-lazy” regime: more specifically, timeline of dynamics ( covered by Lemmas 2-5 )
- Analysis does not require over-parameterization
- A theoretically justified “saddle-to-saddle” behavior conjectured in few related works
- A few interesting supplementary results, e.g., that only for a particular choice of subgradient the dynamics are well-defined.

Weaknesses:
- Orthogonality assumption on data (which ultimately allows for such precise characterization of dynamics). This is the main weak point of the paper, as it is hard to understand where current technique will be of any insight in general (although, authors do a fair job of elaborating on it in the corresponding section)
- A few results in the related literature worth discussing are missing

---

> ### Author Response · Authors · 2022-08-02
> **Answer to Reviewer 4xwj**
>
> Thank you for your careful reading and insightful review of this work. We want to thank the reviewer for pointing out interesting related works, which we will discuss in the revised version. We answer below your specific question.
>
>     1. Authors should be a bit more explicit about stating that the result holds for a particular sequence...
>
> In the case of linear regression (and linearized NTK as in [Yaoyu Zhang, 2020]), it is indeed known that for any initialisation, the parameters converge towards the interpolator that is the closest (in $\ell_2$ norm) to the initialisation. Characterising the exact limit point for any initialisation becomes much more intricate when the model is not linear. For one-hidden layer diagonal networks, the limit point for intermediate initialisations is known but highly non-trivial.
> In harder settings such as matrix factorisation, results on the limit points are only available in the limit $\lambda \to 0$. Similarly, this work only describes the convergence point in the limit $\lambda \to 0$. In this limit, this can be seen as the closest interpolator to the initialisation (which is then $0$). For any non-zero initialisation, we are yet unable to characterise the limit point (we only have a characterisation up to some $\lambda$ scaling term).
> The selected min-norm solution is indeed non-unique as some transformations (e.g. neurons permutation) preserve the parameters norm and training loss. However, all minimal norm interpolators (in parameters space) yield the same estimated function. This function is actually the (unique) minimal $\mathcal{F}_1$ norm interpolator (in function space). Furthermore, some characterisation of the obtained min-norm solution $\theta^*$ is given in the proof of Lemma 12 in the Appendix.
>
> These points will be made clearer in the revised version.

---

### Official Review · Reviewer_7CA5 · 2022-07-10

**Rating:** 7
**Confidence:** 3
**Soundness:** 4 excellent
**Presentation:** 4 excellent
**Contribution:** 3 good

**Summary:**

This paper provides a detailed and quantitative characterization of the gradient flow dynamics of a one-hidden-layer ReLU network on orthogonal data. The results require some additional assumptions: square loss function, regression task, balanced and small initialization, and the width is large enough. Under these assumptions, the authors showed that the gradient flow process can be divided into 4 phases and characterized each phase precisely. The implications of the characterizations of this training process include: training loss will converge to zero, the weights will converge towards some minimum $\ell_2$-norm solution, and the training trajectory will encounter a non-zero strict saddle point. The authors also did experiments to validate their theoretical results.

**Questions:**

- As mentioned in the "weaknesses" section above, it might be better if the authors could give more theoretical intuitions or empirical observations/evidence about neural network training dynamics in more general settings. It could also be better if the authors could provide quantitative requirements and more discussions about the initialization and width of the network.

- Are there more implications for neural network training in practical settings that could be provided by the results in this paper? It would be very interesting if these results could provide guidance on training neural networks in real-world applications, and currently this connection seems a little weak.

**Limitations:**

This paper is mostly theoretical and focuses on very fundamental problems in neural network optimization, and I do not see any immediate negative societal impact of this work. The authors also adequately addressed the limitations of this paper in Section 3.2.

**Strengths And Weaknesses:**

Strengths:

- This paper studies an important problem in this field, i.e., the training dynamics of neural networks. This work characterized the training dynamics in detail in a simplified setting (gradient flow of regression task with one-hidden-layer wide ReLU network, orthogonal input, square loss, and small and balanced initialization), and this dynamics has several indications which align with previous works' conjectures or empirical observations, including global convergence, implicit bias towards minimum $\ell_2$-norm solution, and saddle to saddle dynamics.

- This paper is well-written and well-structured, making it easy to read. The notations are well-defined, and the intuitions behind the lemmas and the implications of the theoretical results are discussed in detail. Besides, the authors provided figures (especially Figure 1) to illustrate the training process, which could help with the readers' understanding of the theoretical results.

- The authors did experiments on synthetic orthogonal data to validate their theoretical results. They also did experiments for non-orthogonal cases, which provided intuitions about the similarities and differences between orthogonal and non-orthogonal settings.

- The theoretical proofs in this paper appear to be correct, and the related works are adequately cited. Besides, the experimental settings are provided in detail, making them easily reproducible.

Weaknesses:

- The input orthogonality requirement might be a bit strong, and it limits the number of training data to be no more than the input dimension. Although the authors did some experiments on non-orthogonal inputs, the training dynamics of neural networks still seem a bit unclear and hard to characterize. It might be better if the authors could provide more intuitions/observations in more general cases.

- The requirements for $\lambda^*$ and the width of the network are not explicitly stated and discussed in the main text, which might make it hard to understand the results quantitatively. It could be better if the authors could move the upper bound of $\lambda^*$ to the main text, and discuss more about the network width requirement. For instance, if the weights are initialized i.i.d. from the unit sphere, we might need exponentially (in the number of training data) many neurons to ensure that Assumption 3 is true with high probability.

Typos:

- In Figure 1, the end of Early Alignment phase should be $-\frac{\epsilon\ln(\lambda)}{||D_-||}$ instead of $-\frac{\epsilon\ln(\lambda)}{||D_+||}$ according to Lemma 2.

---

> ### Author Response · Authors · 2022-08-02
> **Answer to Reviewer 7CA5**
>
> Thank you for your careful reading and insightful review of this work. We here answer in detail your different concerns/questions.
>
>     1. The input orthogonality requirement might be a bit strong...
>
> The general case indeed remains a large mystery and this work only aims at giving some understanding of the training dynamics through a simple instance (orthogonal data). As explained in the answer to *Reviewer bKcM*, the following points also hold in the non-orthogonal case for small initialisations:
>  - the early alignment phase holds in the general case and can be proven, extending our analysis of the first phase
>  - the trajectory goes through intermediate saddle points (see [3])
>  - convergence to zero training loss occurs for a large number of neurons $m$ (see e.g. [1])
>
> However, analysing the dynamics in details is much more intricate. This difficulty mainly comes from the fact that clusters of neurons can simultaneously grow in norm and change direction. This is not the case in the orthogonal setting, where neurons keep a fixed direction while they grow in norm (hence simplifying the analysis). This phenomenon is illustrated by Figure 4.d). We will also make available online animated versions of these figures which will help the understanding of this phenomenon.
>
> Also, we refer to our answer to *Reviewer bKcM* for experiments in the general setting (with high dimensional data) that will be added in the next version.
>
>     2. It could also be better if the authors could provide quantitative requirements and more discussions about the initialization and width of the network.
>
> We refer to our answer to *Reviewer XLXc* for a complete discussion of these points and summarise it here. Assumption 3 indeed needs an exponential number of neurons to hold with high probability for a classical initialisation. Such an exponential number of neurons is actually necessary to have convergence towards minimal norm solution. Only convergence to $0$ loss should hold with fewer neurons (e.g. polynomial in $n$), but the final estimate will not be of minimum norm (but can still yield some sparsity).
>
> The dependency of $\lambda^*$ is of order $\frac{\Theta(1)}{\sqrt{m}}e^{-\Theta(n)}$. While the exponential dependency in $n$ is rather strong, it is consistent with what is generally obtained in the implicit bias literature. Also, this exponential dependency is necessary to observe a separation between phases 2 and 3 (i.e. to observe an intermediate saddle), but the final solution might be similar for larger scales of initialisation (not in the lazy regime).
>
>     3. Are there more implications for neural network training in practical settings that could be provided by the results in this paper?
>
> Our theoretical results mostly aim at explaining and confirming phenomena that have been observed in practice. Some simple key messages can also be kept from this work:
>  - omnidirectional initialisations are crucial to guarantee convergence towards minimum norm interpolator. This type of condition is used in previous theoretical works (see e.g. [1,2]). We here precisely recast how this assumption is required through Assumption 3 (which we recall is a necessary and sufficient condition);
>  - the right scale of initialisation seems to be $\frac{1}{\sqrt{m}}$, as given by the value of $\lambda^*$, which corresponds to the mean field regime;
>  - the number of neurons required to get convergence towards minimal norm solution is exponential in the number of data points. However, we can still have $0$ training loss with some notion of sparsity in the estimated function for fewer neurons as explained in our answer to *Reviewer XLXc*. This provides a new insight on the overparameterisation level needed for $0$ training loss/minimal norm solution.
>
> The first two points correspond to initialisations used in practice (e.g. Xavier or LeCun's).
>
> -----------------------
> [1] Chizat, Lenaic, and Francis Bach. “On the Global Convergence of Gradient Descent for Over-Parameterized Models Using Optimal Transport.” ArXiv:1805.09545, October 29, 2018.
>
> [2] Wojtowytsch, Stephan. “On the Convergence of Gradient Descent Training for Two-Layer ReLU-Networks in the Mean Field Regime.” ArXiv:2005.13530, May 27, 2020.
>
> [3] Jacot, Arthur, François Ged, Berfin Şimşek, Clément Hongler, and Franck Gabriel. “Saddle-to-Saddle Dynamics in Deep Linear Networks: Small Initialization Training, Symmetry, and Sparsity.” ArXiv:2106.15933, January 31, 2022.

---

> > ### Comment · Reviewer_7CA5 · 2022-08-09
> > **Update after authors' response**
> >
> > Thank the authors for the detailed answers to my questions! I have also read all other reviews and the authors' responses, and would like to keep my score unchanged. I would recommend this paper be accepted.

---

### Official Review · Reviewer_XLXc · 2022-07-11

**Rating:** 6
**Confidence:** 4
**Soundness:** 3 good
**Presentation:** 3 good
**Contribution:** 3 good

**Summary:**

This paper studies gradient flow dynamics for two-layer ReLU network with square loss and orthogonal inputs. In particular, authors showed that using small initialization gradient flow will converge to the minimum variation norm interpolator. The theoretical analysis consists of complete training dynamics analysis and highlights an interesting saddle to saddle dynamic. Experiment results are also provided to support the theoretical results.

**Questions:**

I would appreciate if the following questions could be answered:

1. I was wondering if the Assumption 3 can be weakened in some settings so that the required width of neural network can only be polynomial in relevant parameters.

2. I was wondering the exact requirement on \lambda^*, which I did not find in Lemma 12 (line 149)


**Limitations:**

The limitations of the work are clearly discussed. This is a theoretical work thus I don’t see any potential negative social impact.

**Strengths And Weaknesses:**

Originality
1. This work considered the regression setting with small initialization and orthogonal inputs and showed that gradient flow is biased towards minimum variation norm. This result is different from previous implicit bias results and gives understanding of implicit regularization under new settings.

Quality
1. The quality of this paper is overall good. The results and limitations are clearly discussed. Proof sketch of the main result is provided. Proofs of the theoretical results are included in the appendix.

Clarity
1. The paper is overall well-written and easy-to-follow.

Significance
1. Understanding implicit regularization of gradient flow/gradient descent is an important and active research problem towards understanding overparametrized neural networks.
2. Analyzing the whole training dynamics and showing its convergence to a minimum norm interpolator is not an easy work, even in such simplified setting. It is great for authors to prove such a result.
3. The condition on width of neural networks seems need to be exponential in number of training samples to satisfy Assumption 3, when using the gaussian random initialization. It would be better if authors could make it clear when discussing Assumption 3.
4. The condition on the scale of the initialization also seems need to be exponentially small. I would not view this as a major limitation, since using an exponentially small initialization is common in the implicit bias literature.

---

> ### Author Response · Authors · 2022-08-02
> **Answer to Reviewer Reviewer XLXc**
>
> Thank you for your careful reading and insightful review of this work. We here answer in detail your different questions.
>
>     1. I was wondering if the Assumption 3 can be weakened in some settings so that the required width of neural network can only be polynomial in relevant parameters.
>
> We stated Assumption 3 as a deterministic condition, to avoid any requirement on the width $m$ of the network. As observed by several reviewers, the width needs to be exponential in the number of data points $n$ for Assumption 3 to hold with high probability (for isotropic initialisation). We will put emphasis on the role of isotropic initialisation, stating that Assumption 3 is actually a **necessary condition** for convergence towards a minimal norm interpolator.
>
> With a polynomial (in $n$) number of neurons, Assumption 3 does not hold and the analysis becomes very different. In that case the training loss should still converge to $0$, but the $\ell_2$ norm of the parameters will not be minimal. More precisely, the estimated function will have more than $2$ kinks. However, we believe that an adapted analysis might still show some sparsity in the number of kinks of the final estimation. Moreover, the training trajectory should go through more intermediate saddle points, since a new saddle should be visited each time a kink is *created*.
>
> Note also that the orthogonal case might be the worst case in the required number of neurons, as the activation of a neuron (i.e. the data points with which it is positively correlated) cannot change during the training. Assumption 3 thus requires that some cone is not empty at initialisation, since it would remain empty forever otherwise. In the general case, such a cone might be empty at initialisation and gain some neurons during the training. This would make the assumption on the initialisation weaker (in terms of number of required neurons) in the general case. Overall, a precise analysis of these claims is an interesting route for future work!
>
>     2. I was wondering the exact requirement on $\lambda^*$, which I did not find in Lemma 12 (line 149)
>
> The exact value of $\lambda^*$ was indeed omitted for exposition's clarity. However, it is indeed important to stress that our analysis leads to $\lambda^*$ scaling as $\frac{\Theta(1)}{\sqrt{m}}e^{-\Theta(n)}$. Interestingly, the $\frac{1}{\sqrt{m}}$ term is reminiscent of the mean field regime, while the exponential dependency in $n$ is common in the implicit bias literature [Woodworth, 2020]. For larger values of $\lambda$ (but still in the mean field regime), the parameters empirically seem to still converge in a $\mathcal{O}(\lambda)$ vicinity of the minimal norm interpolator. The analysis is yet more intricate in that case; and we do not observe any separation between Phase 2 and Phase 3, i.e. there is no intermediate saddle in the trajectory.

---

### Official Review · Reviewer_bKcM · 2022-07-15

**Rating:** 7
**Confidence:** 3
**Soundness:** 3 good
**Presentation:** 3 good
**Contribution:** 3 good

**Summary:**

This paper inscribes in the direction of providing quantitative understanding on the dynamics and implicit bias of gradient-based optimisation. The authors focus on regression problems in architectures with one-hidden-layer of finite width, ReLU activations and mean-squared error loss. The inputs are fixed and assumed to be pairwise orthogonal. The results are non-asymptotic, holding for any input dimension and training set size.
The contribution is threefold. First, they show global convergence of gradient flow to zero training loss if initialised at sufficiently small norm.  Second, they show an implicit bias of gradient flow towards minimum $\ell_2-$norm interpolator again when starting from sufficiently small initialisation norm. Third, they characterise four different training phases traversed by the dynamics: (i) initial alignment of the neurons along one of two key directions while keeping a small norm, (ii) positively-aligned neurons grow in norm until fitting the positive labels, while the other do not move, (iii) the same for negatively aligned neurons until fitting negative labels, (iv) final convergence. The orthogonality assumption on the inputs is crucial to allow this separation between alignment and norm-growth phases. The authors also show that a saddle-to-saddle dynamics takes place, starting close to the saddle point at the origin and getting trapped for a long time close to a saddle point between phases (ii) and (iii).
The theoretical results are confirmed on a toy dataset of 2 unidimensional data points with a bias.



**Questions:**

I would be interested if the authors could discuss the following points:

a - What are the main limitations of extending the present analysis to the finite-step-size case and multi-pass stochastic gradient descent?

b - What are the main limitations of extending the present analysis to generic losses other than the square one?

In [1], the authors discuss the necessity of an initial alignment with the target function for gradient descent to converge to a solution. Could you comment on how to reconcile the findings in [1] with the result of this paper that GD dynamics can align with the relevant directions? It seems to me that the main discrepancy is again related to the pairwise orthogonality assumption that limits the training set size, which prevents to reconcile the present setting with the dynamics of population loss. Could the authors comment more on this point?


Reference:

[1] Emmanuel Abbe, Elisabetta Cornacchia, Jan Hazla, Christopher Marquis Proceedings of the 39th International Conference on Machine Learning, PMLR 162:33-52, 2022.

**Limitations:**

The main limitations of the work regard the restrictiveness of the assumptions to pairwise orthogonal data and fixed dataset, which prevents from obtaining an understanding of the generalisation properties of the GD algorithm.
The main weakness in my opinion is the experimental setup, which could be improved in the ways I have described above.

**Strengths And Weaknesses:**

Overall I find that the paper is well written. The motivations behind the analysis are relevant and clearly stated in the introduction.
To the best of my knowledge, the analysis of the implicit bias of gradient descent on finite-width one-hidden-layer ReLU networks is novel. Interestingly, the analysis is non asymptotic in the input dimension, dataset size and time.

The assumption of a fixed dataset seems to me a significant limitation towards understanding the generalisation properties in relation to the statistics of the data. It would be interesting if the authors could discuss more about the difficulty of averaging over some given data distribution. The pairwise orthogonality assumption on the inputs is also a significant limitation in terms of the learning ability of the network, given that the number of samples is limited to the input dimensionality.

I also find that the experimental validation of the results is a bit poor and it would be interesting to see the paper findings confirmed on higher dimensional datasets. In particular, it would be interesting to test whether the observed training phenomena still hold true on non-orthogonal but more realistic datasets.

Finally, the manuscript is not completely clear to me in a couple of points:

- The paragraph “Feature learning and small initialisation” (lines 38-45) seems to suggest that the large initialisation case is not interesting since it leads to the lazy training regime. However, it is not clear and should be clarified whether the analysis presented in this paper has been already derived for the precise setting under consideration at large initialisation norm.
- At line 110, the authors write that their gradient flow description holds for the infinitesimal step-size limit of "(stochastic) gradient descent" implying that as the step size vanishes there is no distinction between SGD and GD. If the authors want to keep this comment they should justify it providing a reference or an explanation in support of this claim.

---

> ### Author Response · Authors · 2022-08-02
> **Answer to Reviewer bKcM (1/2)**
>
> Thank you for your careful reading and insightful review of this work. We here answer in detail your different questions.
>
>     1. The assumption of a fixed dataset seems to me a significant limitation towards understanding the generalisation properties.
>
> The aim of the paper is to provide an understanding of the *training* of a one-hidden layer neural network. Hence, as in concrete applications, the dataset is finite and fixed (size $n$). However, beyond the convergence results, the **implicit bias** characterisation anticipates exactly the generalization properties of the returned estimate.
> Indeed, as we show that the  retrieved function is of minimal $\mathcal{F}_1$ norm (equation (8) page 5), good guarantees on the population loss can be derived when the dataset is sampled i.i.d. from a population distribution [Kurková and Sanguineti, 2001 and Bach, 2017]. We will make these crucial facts clearer in the revision.
>
>
>     2. The pairwise orthogonality assumption on the inputs is also a significant limitation in terms of the learning ability of the network.
>
> If the detailed characterisation of the dynamics in the non-orthogonal case indeed remains a mystery, we believe that the present analysis can shed light on an important and more general phenomenon. Indeed, at small initialisation, we conjecture that the **simplicity bias** towards low $\mathcal{F}_1$-norm is expected to hold beyond the orthogonal setup, as observed experimentally in Figure 4.e). From this, as explained in the previous paragraph, good generalisation properties of neural networks can be directly shown.
>
>     3.  I also find that the experimental validation of the results is a bit poor and it would be interesting to see the paper findings confirmed on higher dimensional datasets. In particular, it would be interesting to test whether the observed training phenomena still hold true on non-orthogonal but more realistic datasets.
>
> We agree with the reviewer and will add experiments in higher dimensional settings. As mentioned in the paper, concentration properties inherent to the high-dimension (large $d$) imply that the data inputs might be almost orthogonal: this is for example the case for i.i.d. standard Gaussian variables. We will complement our paper with experiments showing that the exact same phenomena occur in this case: the dynamics travels near a unique intermediate saddle point and show the same simplicity bias. For more realistic dataset, even if the story may be more complicated, some of the mentioned phenomena will still occur:
>  - the early alignment phase holds in the general case and can be proven, extending our analysis of the first phase
>  - the trajectory goes through intermediate saddle points (see [3])
>  - convergence to zero training loss occurs for a large number of neurons $m$ (see e.g. [2])
>
> We refer to the mentioned references for experimental evidence of the described phenomena. However, fully characterising the simplicity bias for realistic datasets remains an interesting and open question.
>
>     4.  Suggestion that the large initialisation case is not interesting since it leads to the lazy training regime.
>
> We do not want to say that the large initialisation case is not interesting. However, in this case, the problem has already  been fully solved:  convergence and implicit bias are well-documented in the literature as the dynamics is  the same as the one of a linear method on the Neural Tangent Kernel features [Jacot et al, 2018]. Note furthermore that orthogonality would not be required and would not bring any rich particularities in this linear regime. Let us also stress that the large initialisation regime is known  not to be as **rich** as the feature learning regime we study in our paper. Indeed, it has been proven, e.g. [Bach, 2017], that the latter has better generalisation properties in practically-relevant settings.
>
>     5. No distinction between SGD and GD.
>
> We will add the reference [1], where it is proven that at first order in the step-size there is no distinction between SGD and GD.
>
>      6. Finite step size GD and multipass SGD.
>
> The main difficulty in handling finite step size or analyzing multipass SGD lies in the fact that the neurons could change of sectors (i.e., level sets of their activation vector). In fact these changes correspond to the locations of the discontinuity of the ReLu gradients and hence the non-smoothness of the training loss. This interesting technical extension is left for future work. Note finally that due to the difficulty of understanding neural networks dynamics, most of the recent papers on the topic restrict their analysis to continuous gradient flows.

---

> > ### Author Response · Authors · 2022-08-02
> > **Answer to Reviewer bKcM (2/2)**
> >
> >     7. Beyond square loss.
> >
> > Square and logistic losses are the most prominent choices to train neural networks. In the article, we mention how logistic loss has already been studied, see e.g. [Lyu, 2021] and we study the square loss to complete the picture. The fine grained description we provide is very specific to the square loss studied and even if the technical tools we developed may be reused for other losses, we believe there is no universality in these non-convex gradient flows.
> >
> >     8. Relationship with An Initial Alignment between Neural Network and Target is Needed for Gradient Descent to Learn.
> >
> > We believe that there is no incompatibility between our work and the mentioned paper.
> > Abbe et. al provide an impossibility result stating that some initial alignment condition is necessary to learn the target function for any algorithm. On the other hand, our work provides a positive result where gradient descent converges towards a minimal norm solution of the objective.
> >
> > These results are *reconcilable* since the definitions of alignment considered in both works differ. Their initial alignment condition (see their Definition 2.2) states that some neurons are positively correlated with the target function at initialisation. This condition is verified in our setting as soon as the initialization distribution is omnidirectional. Our Assumption 3 is actually a stronger version of this condition (stronger is required for the implicit bias). The initial correlation of the neurons is then reinforced by the early alignment phase, leading to our main result.
> >
> > Note furthermore that the setup of the mentioned paper is very different from ours, in particular, the authors deal with a specific definition of learnability (see Definition 2.3 -Weak learning-) and with functions on the Boolean Hypercube.
> >
> >
> > -----------------
> >
> > [1] Li, Q., Tai, C. and Weinan, E. Stochastic modified equations and dynamics of stochastic gradient algorithms i: Mathematical foundations. The Journal of Machine Learning Research, 20(1), 1474-1520, 2019.
> >
> > [2] Chizat, Lenaic, and Francis Bach. “On the Global Convergence of Gradient Descent for Over-Parameterized Models Using Optimal Transport.” ArXiv:1805.09545, October 29, 2018.
> >
> > [3] Jacot, Arthur, François Ged, Berfin Şimşek, Clément Hongler, and Franck Gabriel. “Saddle-to-Saddle Dynamics in Deep Linear Networks: Small Initialization Training, Symmetry, and Sparsity.” ArXiv:2106.15933, January 31, 2022.

---

> > > ### Comment · Reviewer_bKcM · 2022-08-08
> > > **Answer to authors**
> > >
> > > I thank the authors for the clarifications. After taking the time to read through your comments, I have decided to raise my score since I found them satisfactory and I believe the paper is solid and worth publication.

---

### Meta-Review · Area_Chair_wJAA · 2022-08-20

**Recommendation:** Accept
**Confidence:** Certain

**Metareview:**

This paper studies the gradient flow dynamics for a one-hidden layer ReLU network with square loss and orthogonal inputs. In particular, the authors show that gradient flow converges towards a global minimum for sufficiently small initialization and that the solution consists in the minimum $\ell_2$ norm interpolator. Interesting insights are provided on the various phases of the dynamics, which is characterized by an initial alignment, followed by fitting the positive labels, then the negative ones and a final convergence. A saddle to saddle dynamics (conjectured in earlier work) is also unveiled.

The assumption of orthogonal inputs is rather strict, and it is the main weakness of the paper. That being said, the reviewers and this AC agree that the results of this paper are novel and interesting for the NeurIPS community. Hence, I am happy to recommend acceptance. As a final note, I would like to encourage the authors to include in the camera ready the discussions related to the feedback received from the reviewers (in particular, the experiments in higher dimensional settings).


**Award:**

No

---

### Decision · Program_Chairs · 2022-09-14

Accept